# Adversarial attacks against supervised machine learning based network intrusion detection systems

**Ebtihaj Alshahrani**[1]*, **Daniyal Alghazzawi**[1], **Reem Alotaibi**[2], **Osama Rabie**[1]

**1** Information Systems Department, Faculty of Computing and Information Technology, King Abdulaziz University, Jeddah, Saudi Arabia, **2** Information Technology Department, Faculty of Computing and Information Technology, King Abdulaziz University, Jeddah, Saudi Arabia

* ealshahrany0001@stu.kau.edu.sa

**Data Availability Statement:** The relevant data is publicly available at: https://www.unb.ca/cic/datasets/ids-2017.html. The code may be accessed

## Abstract

Adversarial machine learning is a recent area of study that explores both adversarial attack strategy and detection systems of adversarial attacks, which are inputs specially crafted to outwit the classification of detection systems or disrupt the training process of detection systems. In this research, we performed two adversarial attack scenarios, we used a Generative Adversarial Network (GAN) to generate synthetic intrusion traffic to test the influence of these attacks on the accuracy of machine learning-based Intrusion Detection Systems (IDSs). We conducted two experiments on adversarial attacks including poisoning and evasion attacks on two different types of machine learning models: Decision Tree and Logistic Regression. The performance of implemented adversarial attack scenarios was evaluated using the CICIDS2017 dataset. Also, it was based on a comparison of the accuracy of machine learning-based IDS before and after attacks. The results show that the proposed evasion attacks reduced the testing accuracy of both network intrusion detection systems models (NIDS). That illustrates our evasion attack scenario negatively affected the accuracy of machine learning-based network intrusion detection systems, whereas the decision tree model was more affected than logistic regression. Furthermore, our poisoning attack scenario disrupted the training process of machine learning-based NIDS, whereas the logistic regression model was more affected than the decision tree.

## 1 Introduction

Artificial intelligence has a wide range of applications, especially in the field of cybersecurity, and it is likely to be used for a variety of defensive and offensive purposes [1]. One disadvantage of using artificial intelligence methods to conduct classifications is the potential of attackers attempting to defeat the classifiers by exploring for vulnerabilities in the model's classification process.

Adversarial Machine Learning (AML) is the field that explores and analyzes these types of attacks [2]. The majority of adversarial machine learning research has concentrated on the

at: https://github.com/Ebtehaj431-hub/Adversarial_AI.git.

**Funding:** The Deanship of Scientific Research (DSR) at King Abdulaziz University (KAU), Jeddah, Saudi Arabia has funded this project, under grant no. (IFPDP-289-22). The funders had no role in study design, data collection and analysis, decision to publish, or preparation of the manuscript.

computer vision domain, notably picture recognition, due to the availability and ease of use of well-known datasets [3]. However, adversarial attack generation approaches have been extensively used in various domains in recent years, including natural language processing [4]. However, so many of these attacks have also been applied in the field of cybersecurity [5].

The cybersecurity field is particularly important since it is teeming with adversaries, for example, malware developers attempting to avoid machine and deep learning-based next-generation anti-virus technologies, spam filters, intrusion detection systems, and so on. The adversarial examples are data samples that have been modified by adversaries to exploit learned patterns in the machine and deep learning models, causing the model to misclassify generated samples [6]. This type of attack method is called an evasion attack. The data source can also be a threat, since an adversary can manipulate the data used to train the classifiers, causing the training process to degrade and getting access to adversarial examples. This type of attack method is called a poisoning attack. The artificial intelligence-based-defensive applications are helping to monitor the system if there is any suspicious behavior.

An IDS is one of the security mechanisms' applications which adopts artificial intelligence for tracking traffic passing on networks and through systems to check for suspicious activity and known threats, sending warnings when such objects are detected [7]. The alerting data would provide information about the intrusion's source address, the target/victim address, and the type of attack that is suspected. Typically, the IDS indicates the kind of weakness an attacker is attempting to exploit.

In this era, the attackers become smarter to discover new ways to create new attacks using emerged technologies to accomplish tasks that would be impractical for humans such as machine learning and deep learning. Thus, the malicious use of artificial intelligence results in increased security threats that require the defender to work smarter to predict, prevent, and mitigate these types of threats [8]. Furthermore, the accuracy rate of the IDS model that uses AI technology is not at the acceptance level for detecting zero-day attacks. During the development of the detection model's accuracy rate, the main limitation of IDSs is the limitation of increasing the false-negative rate that indicates undetected intrusions [9].

Most of the IDS models use real network traffic to create a detection model that detects potential similar attacks which may not be enough to improve the accuracy rate of the detection model. Thus, to build high-quality classifiers, it is important to understand and simulate the attack behavior, and study the ways of improving the performance of classifiers to be more resistant against adversarial attacks. One method that can be used to improve the performance of classifiers is developing models that are more resistant to these types of attacks [10].

The main objectives of this research are: first, to investigate and analyze potential NIDS attacks, and second, to investigate the performance of NIDS learning models and the possibility of improving them to make them more resistant to adversarial attacks. The third objective is to illustrate how to perform both evasion and poisoning attacks against the intrusion detection systems dataset by generating a synthetic intrusion network to simulate the future attack that can be passed through the network to keep up with hackers.

**As per the title of this research, three Research Questions (RQs) are developed to conduct our research:**

- Can we generate synthetic intrusion traffics by Deep Learning algorithms?

- Does the generated network traffic by GAN defeat the accuracy of the machine-learning detection model?

- How will the generated data impact the detection rate of the machine-learning detection model as evasion and poisoning attacks?

The remainder of this paper is organized as following: In section 2, we clarified the main concepts in our field of study. In section 3, literature review was discussed about our topic. In section 4, the preparation sittings and details of our experiments were explained. In section 5, the results of experiments were discussed. Finally, in section 6, we summarized the outcome of this research.

## 2 Background

In this section, the major concepts of the work related to this research will be presented. Then, we provide context for the architecture and classification of IDS. In addition, explanations of key concepts about adversarial attacks will be provided.

There are two types of intrusion detection systems: host or network-based intrusion detection systems: [11]:

### 2.1 Host Based Intrusion Detection systems

The Host-Based Intrusion Detection systems (HIDSs) comprise of agent programs run on the monitored devices, in order to work on data gathered from within the computer, primarily operating system audit trails and system logs. The primary benefit of HIDS is the ability to observe actions on the real target, which provides more conclusive proof that an assault is occurring. HIDSs, however, have a few drawbacks. The monitored computer's computational resources are used by HIDSs. A population of computers needs to be covered by numerous HIDSs; one HIDS only covers a single host. Also, they share the same danger as the host who is being attacked. Additionally, the HIDS may be disabled or unreliable if the host is hacked.

### 2.2 Network based Intrusion Detection System classifications

With the growth of potential threats on the internet, Network Intrusion Detection Systems (NIDSs) have become a critical tool for detecting and defending against network attacks carried by malicious network traffic. The IDS analyzes the network traffics and sends an alarm if it detects potentially harmful traffic. The fundamental goal of NIDS is to distinguish between legitimate and illegitimate network records.

**There are two types of IDS strategies**: [12, 13]

- Signature-based Network Intrusion Detection Systems: The detection is based on prior knowledge of the types of attack and the rule identified by the administrator. By comparing observed activity to defined patterns, such systems can classify known attacks, but they can't detect zero-day attacks.

- Anomaly-based Network Intrusion Detection Systems: It detects abnormalities by learning the features of normal activity and then looking for deviations from the normal features. It is capable of detecting new attacks, particularly Zero-Day attacks, although it has a high false alarm rate.

**2.2.1 Anomaly-based network intrusion detection systems.** In anomaly-based network intrusion detection systems, machine learning and deep learning techniques have been extensively used with good outcomes. According to this survey [14], the ML algorithms that are commonly used for Network Intrusion Detection Systems are Decision Tree (DT), K-Nearest Neighbor (KNN), Artificial Neural Network (ANN), Support Vector Machine (SVM), K-Mean Clustering, Fast Learning Network, and Ensemble Methods. The DL algorithms that are commonly used for IDS are Recurrent Neural Network (RNN), Autoencoder (AE), Deep

Neural Network (DNN), Deep Belief Network (DBN), and Convolutional Neural Network (CNN) [14]. We worked on Decision Tree and Logistic Regressions models, and they are some of the algorithms used for Network Intrusion Detection.

**2.2.2 Decision tree.**    A Decision Tree (DT) is a rule-based tree-structured classification model that is learned using all features in the training data to obtain information. The tree structure of this method includes nodes, branches, and leaves. Each leaf represents a prediction result or category label, while the branch represents a rule [15]. To minimize over-fitting, the DT algorithm identifies the optimal features for forming a tree and then prunes the tree to remove unnecessary branches.

**2.2.3 Logistic regression.**    LR is a supervised classifier that uses a discriminative model when training on data. The input features are considered to be distributed a priori in LR models. The amount of the training data has a significant impact on their performance [16].

## 3 Literature review

After the adversarial examples were developed in 2014 by Szegedy et al [17], some of the research has been conducted in all domains wherein machine learning and artificial intelligence have been found an application. Grosse et al [18] presented the first application of generating adversarial examples in the cybersecurity domain. They demonstrated how adversarial examples may be used to exploit a malware detection system based on a neural network and demonstrated how hackers can use this strategy to mask attack activities.

Rigaki and Elragal [19] were the first researchers in testing the effectiveness of adversarial attacks in an intrusion detection scenario. The authors implemented an evasion attack scenario to test the efficacy of adversarial attacks against NIDS based on machine learning and to study the robustness of the most common classifiers based on different metrics under adversarial attacks. This research used FGSM and JSMA strategies for generated adversarial examples and used five learning models as Intrusion Detection Systems: DT, RF, Support Vector Machine (SVM), Voting ensemble, and MLP. The adversaries were supposed to know the features of the target classifier, and one of the tactics modified so many of them. The results showed that the overall models reduced IDS classifier detection accuracy from 5% to 28%.

Some research studies conducted the performance of adversarial attacks as well as the performance of target NIDS against these attacks simultaneously. For example, in [20, 21], authors targeted machine learning-based NIDS, whereas, in those research papers [22], they targeted deep learning-based NIDS. Besides, there are some research studies [23–25] that aimed to study the effect of this type of attack by various attack strategies on different types of NIDS.

Most of the existing research papers on adversarial attacks focused on and studied the evasion attacks against various learning models of NIDS, which is a type of adversarial attack. Notably, the poisoning attack is a type of adversarial attack, which has not been studied enough in existing research papers. There are just two research that studied and discussed the potential impact of poisoning attacks against ML-based NIDS and DL-based NIDS [20, 26].

The authors [20] studied different attack scenarios in three areas (intrusion detection systems, spam detection, malware analysis) and test the countermeasures of classifiers before and after adversarial attacks. They focused on defender perspectives more than attackers against adversarial attacks. In the other study [26], the authors provided a new method of poisoning machine learning-based IDSs by stealing specific targeted models. To offer suitable data for training substitute models using DNNs, they first develop an improved synthetic data generating technique called A-SMOTE. The next phase entails providing a powerful poisoning technique called CBPC, which can create poisoning samples using the knowledge from augmented training data. The final stage is to describe a poisoning method

that can use poisoned samples created by simulating a poisoning attack against substitute models to poison the targeted models.

In this research, we are extensively studying evasion and poisoning attacks and analyzing their impacts on the real models of NIDS. For two purposes: first, investigate and analyze potential adversarial attacks against NIDS; second, investigate the performance of NIDS learning models against these attacks.

## 4 Experiments

All experiments were conducted using Python 3.8. Machine learning libraries were required to develop the implementation of numerous techniques easier. Pandas [27], NumPy [28], Matplotlib [29], and Sklearn [30] were used for preprocessing operations and data representation. We useed the scikit-learn library to implement the Decision Tree, Logistic Regression and its performance metrics.

Furthermore, we generated synthetic data using the TensorFlow and Random libraries with the GAN model.

### 4.1 Dataset and preprocessing

The CICIDS2017 dataset was used based on selection properties mentioned in [31]. The CICIDS2017 dataset consists of about 2800000 records of network traffic in packet-based and flow-based formats that were captured by the emulated network environment. The dataset was captured in 5 days. It captured some attack scenarios such as Secure Socket Shell (SSH), brute force, heartbeat, botnet, denial of services (DoS), distributed denial of services (DDoS), web, and infiltration attacks within their data set from real network traffic. The dataset consists of 15 classes of data: SSH brute force, heartbeat, botnet, DDoS, DDoS, web, and infiltration attacks. It consists of 83% of the benign samples and 17% of the attack samples. The details of the statistical description of the dataset are illustrated in the following Table 1.

Several preprocessing operations were required before the machine learning models were trained and tested on the datasets. The preprocessing operations included deleting the missing and infinite values, re-grouping the dataset by type of attacks, and labelling encoder and power transformer for normalizing the features of datasets.

### 4.2 Adversarial attack technique

To perform IDS detection, we chose to implement two types of machine learning models, Decision Tree, and Logistic Regression, as our attack targets. These two algorithms are the most used classifiers in IDS [14].

We used the Generative Adversarial Network strategy to generate the synthetic dataset that would help in mimicking the adversarial attack against NIDS. We used GAN to generate

**Table 1. Statistical description of the dataset.**

| Name | Value |
|---|---|
| Rows | 2830743 |
| Columns | 80 |
| Discrete Columns | 1 |
| Continuous Columns | 79 |
| Missing Data | 1358 |
| Infinite Data | 1509 |
| Memory Allocation | 235.1 MB |

network traffic based on learned characteristics from the original dataset for the following purposes:

- In a variety of areas, such as voice and image recognition systems, the GAN algorithm is the most utilized weapon by attackers.

- The GAN algorithm is considered one of the most promising solutions in artificial intelligence due to its ability to generate data from the few data available through the Generator.

- The GAN algorithm is effective for generating a new set of intrusive data, and it will be employed to disrupt the learning process of AI-based IDS to analyze model weaknesses.

GANs (generative adversarial networks) are a neural network model that outperforms prior generating methods. GAN has two types of neural networks, a generator (G) and a discriminator (D). First, the generator is used to generate a new network traffic dataset based on input datasets. Then, the discriminator is used to distinguish between the generated traffic dataset and origin datasets [32].

GANs generate realistic data by recognizing the structure in the data. GANs have many types of structures such as standard GAN, DCGAN (Deep Convolutional Generative Adversarial Network), CycleGAN, and so on. In this research, we used DCGAN which was first introduced by [33]. A DCGAN is a straightforward extension of the GAN mentioned above, with the exception that the discriminator and generator layers are specifically convolutional and convolutional-transpose layers. The following Fig 1 shows the structure of the DCGAN model that we used.

## 4.3 Experiments' setup

In the first experiment, we used Generator Adversarial Network (GAN) to generate adversarial examples as an adversarial strategy.

- **Data initialization**: The dataset consists of Seven categorical values from benign and types of attacks. It is an unbalanced dataset where the benign data were 83% of original datasets. Due to the amount of some categories being more than others, GAN models couldn't learn some categories that have a few amounts of data. Therefore, we chose almost equal samples from each category to improve GAN's learning of each category. Accordingly, the performance of GAN-generated samples would be enhanced.

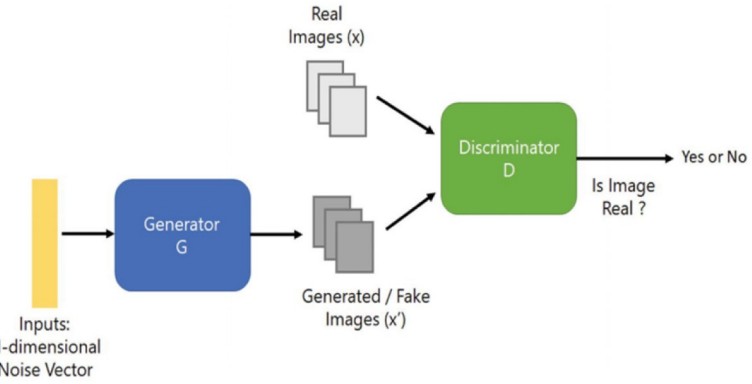

**Fig 1. DCGAN structure** [34].

- **Data splitting**:The GAN model was trained using all the original data, then the model was tested and evaluated using 50,000 generated traffics.

- **Model settings**: A GAN is made up of two neurons, a discriminator, and a generator. All the neurons have one input layer, two hidden layers, and one output layer. The ReLU activation functions were contained in both the generator and the discriminator. The generated dataset was based on random noise with 32 noise dimensions. The following Figs 2 and 3 demonstrator the structures of neurons in detail.

In second experiment, we used the generated data by GAN to perform the evasion attack scenario.

- **Data initialization**: The entire dataset was used as is after preprocessed operational with the same distributions in 7 categories from benign and types of attacks. Even the generated data has been used.

- **Data splitting**: The dataset was split into two parts: 70% for training and 30% for testing the Decision Tree and Logistic Regression models. Additionally, the generated data was used as a part of testing on the target machine learning models which were 50,000.

```
Layer (type)              Output Shape          Param #
=================================================================
input_1 (InputLayer)      [(32, 32)]            0

dense (Dense)             (32, 128)             4224

dense_1 (Dense)           (32, 256)             33024

dense_2 (Dense)           (32, 512)             131584

dense_3 (Dense)           (32, 79)             40527
=================================================================
Total params: 209,359
Trainable params: 209,359
Non-trainable params: 0
```

**Fig 2. Generator network description.**

```
Layer (type)              Output Shape          Param #
=================================================================
input_2 (InputLayer)      [(32, 79)]            0

dense_4 (Dense)           (32, 512)             40960

dropout (Dropout)         (32, 512)             0

dense_5 (Dense)           (32, 256)             131328

dropout_1 (Dropout)       (32, 256)             0

dense_6 (Dense)           (32, 128)             32896

dense_7 (Dense)           (32, 1)               129
=================================================================
Total params: 205,313
Trainable params: 0
Non-trainable params: 205,313
```

**Fig 3. Discriminator network description.**

- **Model settings**: We trained the Decision Tree and Logistic Regression models on the original dataset. Then, the models were tested twice, once on the original dataset and secondly on the original dataset with generated dataset.

   Finally, in the third experiment, we used the generated data by GAN to perform the poisoning attack scenario.

- **Data initialization**: We selected a 5% sample from an original dataset, considering data distribution preservation. The generated data was used with 75% of its labels flipped (Random Flipping). The flipping labels were incorporated randomly between all labels. These procedures were carried out since the number of created data was still insignificant compared to the number of original data, which could reduce the effectiveness of our attack scenario.

- **Data splitting**: The sampled dataset was split into two parts: 70% for training and 30% for testing the Decision Tree and Logistic Regression models. Additionally, the generated dataset was used as a part of the training dataset on the target machine learning models which were 50000.

- **Model settings**: All of the Decision Tree and Logistic Regression models were trained in two ways: first on original sampled dataset, and then on original sampled dataset with poisoned dataset (the generated dataset that has been flipped). In all ways, we tested the models on the testing set of the original sampled dataset.

## 4.4 Performance metrics

The Confusion Matrices, Accuracy, Precision, Recall and F-measure of the target machine learning-based NIDSs were used to measure the performance of our adversarial attacks [35].

- **Confusion matrices**: The confusion matrices are based on the four measurements as shown in Table 2: True Positive (TP), True Negative (TN), False Negative (FN), and False Positive (FP).

  - TP: is the total number of malicious data identified as an attack.

  - FP: is the total number of benign data identified as an attack.

  - TN: is the total number of benign data identified as benign.

  - FN: is the total number of malicious data identified as benign.

- **Accuracy**: Accuracy is at kind of measurement, which is defined as the ratio of correct to total predictions.

$$Accuracy = \frac{CorrectPredictions}{TotalPredictions} \tag{1}$$

**Table 2. Confusion matrices.**

|  | Predicted Positive | Predicted Negative |
|---|---|---|
| **Actual Positive** | Total of True Positive (TP) | Total of False Negative (FN) |
| **Actual Negative** | Total of False Positive (FP) | Total of True Negative (TN) |

- **Precision**: Precision refers to a rate of positive samples from all the samples predicted as positive by classifiers.

$$Precision = \frac{TruePositive}{TruePositive + FalsePositive} \tag{2}$$

- **Recall**: Recall refers to the rate of positive samples that are correctly predicted from all the correctly predicted samples.

$$Recall = \frac{TruePositive}{TruePositive + TrueNegatives} \tag{3}$$

- **F-Measure**: F1 is the weighted average of Precision and Recall, which is constructed using the harmonic mean of Precision and Recall.

$$F1 = 2 * \frac{Precision * Recall}{Precision + Recall} \tag{4}$$

## 5 Results and discussion

Regarding the performance of GAN in generating adversarial examples, the GAN model's training phase was unstable, with training accuracy ranging from 85% to 92%. The training operations used 3,000 epochs with a learning rate of 0.0005. We observed GAN didn't create data from all categories of the dataset during testing and evaluation on 50.000 generated data.

In the following Figs 4 and 5, let us see whether the generated data matches the actual data once we've trained the model. To see how the plot for the generated data altered as the networks learned the embedding more efficiently, we plotted the generated data for some of the model steps.

On the original dataset, we trained Decision Tree and Logistic Regression models. The models were then tested twice: once on the original dataset and again on the original dataset with generated dataset to measure the performance of machine learning models against evasion attacks. The testing results of both models are shown in the Tables 3 and 4:

As we have shown in this attack scenario, the accuracy of machine learning model-based network intrusion detection systems was affected, which means that the machine learning models didn't classify all the generated dataset by GAN even though it was similar to the original dataset. Furthermore, the created dataset exceeded NIDS and reduced the accuracy of both NIDS models. That means our evasion attack scenario negatively affected the accuracy of machine learning-based NIDS, whereas, the decision tree model was more affected than logistic regression in this case.

To test the performance of machine learning models against poisoning attacks, we trained all Decision Tree and Logistic Regression models twice: first on the original sampled dataset, and then on the original sampled dataset with the poisoned dataset (the generated dataset that has been flipped). In all ways, we tested the models on the testing set of the original sampled dataset. In the Tables below 5–8, we presented the results of all models on original data, followed by the results of the models under poisoning attack scenarios.

As we have demonstrated, the poisoning of the training dataset had an impact on the training accuracy of machine learning models, but had a lesser impact on the testing

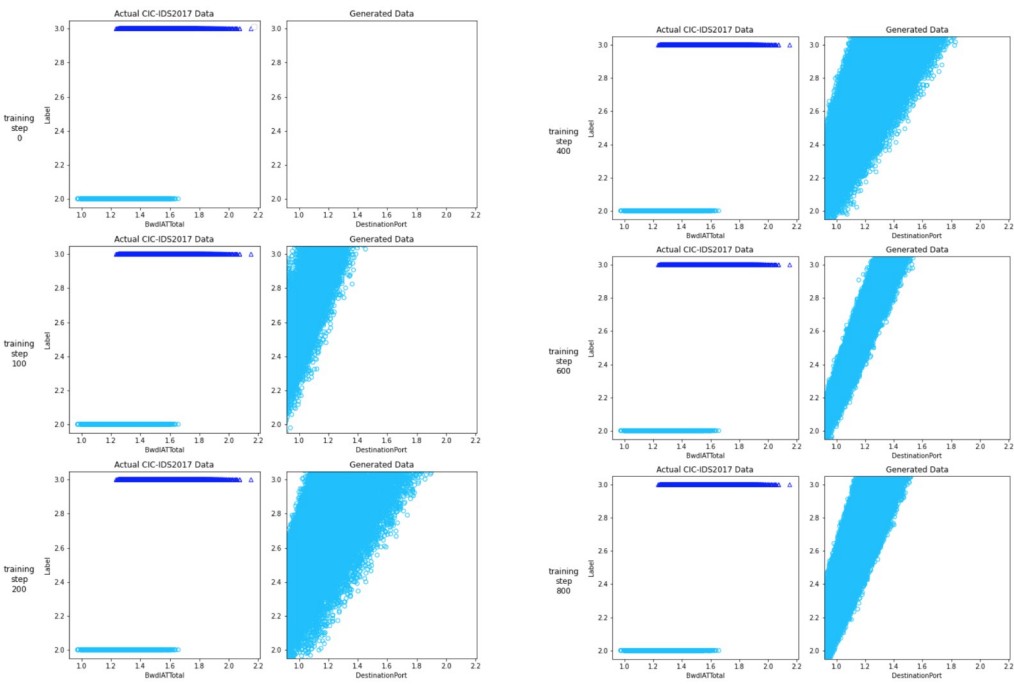

**Fig 4. Comparison of GAN outputs.**

accuracy. This is expected for a variety of reasons, including those machine learning models that were trained on both the original dataset and a poisoned dataset that represents 25% of total training samples. This implies that the poisoning attack was designed to disrupt training processes, regardless of testing process. Furthermore, the poisoning attack scenario

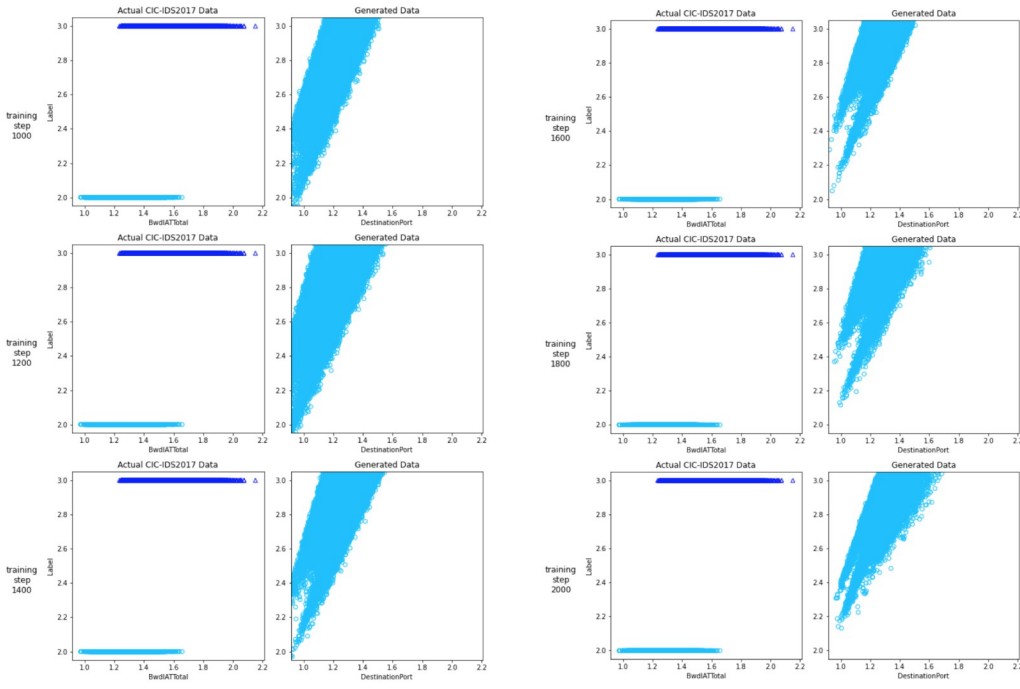

**Fig 5. Comparison of GAN outputs.**

**Table 3. Performance of testing on original dataset.**

| Model | Accuracy | Precision | Recall | F1 |
|---|---|---|---|---|
| DT | 99% | 95% | 93% | 94% |
| LR | 99% | 76% | 62% | 64% |

**Table 4. Performance of testing on original dataset with generated dataset.**

| Model | Accuracy | Precision | Recall | F1 |
|---|---|---|---|---|
| DT | 94% | 81% | 78% | 71% |
| LR | 96% | 74% | 50% | 53% |

**Table 5. Results of training on the original dataset.**

| Model | Accuracy | Precision | Recall | F1 |
|---|---|---|---|---|
| DT | 98% | 65% | 59% | 61% |
| LR | 98.9% | 80% | 73% | 74% |

**Table 6. Results of testing on the original dataset.**

| Model | Accuracy | Precision | Recall | F1 |
|---|---|---|---|---|
| DT | 98% | 55% | 50% | 52% |
| LR | 98.8% | 66% | 61% | 61% |

**Table 7. Results of training under poisoned data.**

| Model | Accuracy | Precision | Recall | F1 |
|---|---|---|---|---|
| DT | 97.6% | 69% | 58% | 60% |
| LR | 95.8% | 67% | 56% | 58% |

**Table 8. Results of testing under poisoned data.**

| Model | Accuracy | Precision | Recall | F1 |
|---|---|---|---|---|
| DT | 97.8% | 55% | 50% | 52% |
| LR | 98% | 55% | 51% | 50% |

affected the test accuracy less because the samples utilized in the testing phase matched some of the training process samples.

We can state that our poisoning attack scenario disrupted the machine learning-based NIDS training process by randomly label flipping 25% of all training data, with the logistic regression model being more affected than the decision tree in this case.

We expected the poisoning attack to have a greater impact than the evasion attack because it aims to interrupt the training process of the target machine learning models. Due to a variety of limitations, this expectation was not realized. We worked on just 5% of the original dataset since the number of created data was few compared to the number of original data, which could reduce the effectiveness of our attack scenario. Furthermore, the flipping function takes a long time when it is working on the dataset.

## 6 Conclusion

We used the GAN model to randomly generate adversarial samples for both evasion and poisoning attack scenarios. These generated samples will be undetected using machine learning classifiers in evasion attacks or will disrupt the training process in poisoning attacks. Experimental work has been performed using the CICIDS2017 dataset. The results show the performance of the proposed evasion and poisoning attacks were affected the accuracy of Decision Tree and Logistic Regression models in performed attack scenarios. Furtherly, we can answer the three research questions given in 1 using the findings of the experiments.

- **Can we generate synthetic intrusion traffics by Deep Learning algorithms?**
  Yes: we used GAN, one of the deep learning algorithms, to generate adversarial samples, with a success rate of 92% in our experiment. This means that it has the ability to generate data from the dataset as an adversarial strategy.

- **Does the generated network traffic by GAN defeat the accuracy of the machine-learning detection model?**
  Yes: as represented in all attack scenarios, the accuracy of both Decision Tree and Logistic Regression classifiers were affected and reached 94% and 96% in the evasion attack scenario, and 97% and 95% in the poisoning attack scenario, respectively.

- **How will the generated data impact the detection rate of the machine-learning detection model as evasion and poisoning attack?**
  The results show that the proposed evasion attacks reduced the testing accuracy of all machine learning models, whereas the poisoning attack affected the training accuracy of machine learning models but not the testing accuracy.

## Author Contributions

**Conceptualization:** Ebtihaj Alshahrani, Daniyal Alghazzawi, Reem Alotaibi.

**Data curation:** Ebtihaj Alshahrani.

**Formal analysis:** Ebtihaj Alshahrani.

**Funding acquisition:** Daniyal Alghazzawi.

**Investigation:** Ebtihaj Alshahrani, Reem Alotaibi.

**Methodology:** Ebtihaj Alshahrani, Daniyal Alghazzawi, Reem Alotaibi.

**Project administration:** Daniyal Alghazzawi, Reem Alotaibi, Osama Rabie.

**Resources:** Ebtihaj Alshahrani.

**Software:** Ebtihaj Alshahrani.

**Supervision:** Daniyal Alghazzawi, Reem Alotaibi, Osama Rabie.

**Validation:** Ebtihaj Alshahrani, Daniyal Alghazzawi, Reem Alotaibi.

**Visualization:** Osama Rabie.

**Writing – original draft:** Ebtihaj Alshahrani.

**Writing – review & editing:** Daniyal Alghazzawi, Reem Alotaibi.

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
