## [Decision Letter · Decision Letter 0]

4 Jul 2022

PONE-D-22-13930Adversarial Attacks Against Supervised Machine Learning Based Network Intrusion Detection SystemsPLOS ONE

Dear Dr. Alshahrani,

Thank you for submitting your manuscript to PLOS ONE. After careful consideration, we feel that it has merit but does not fully meet PLOS ONE’s publication criteria as it currently stands. Therefore, we invite you to submit a revised version of the manuscript that addresses the points raised during the review process.

We look forward to receiving your revised manuscript.

Kind regards,

Dr. Omar A. Alzubi

Academic Editor

PLOS ONE

Journal Requirements:

    "The Deanship of Scientific Research (DSR) at King Abdulaziz University (KAU), Jeddah, Saudi Arabia has funded this project, under grant no. (RG-10-611-43)."

   "The Deanship of Scientific Research (DSR) at King Abdulaziz University (KAU), Jeddah,Saudi Arabia has funded this project, under grant no. (RG-10-611-43)."

    "The Deanship of Scientific Research (DSR) at King Abdulaziz University (KAU), Jeddah, Saudi Arabia has funded this project, under grant no. (RG-10-611-43)."

6. Please ensure that you refer to Figures 1-4 in your text as, if accepted, production will need this reference to link the reader to the figure.

7. We note you have included a table to which you do not refer in the text of your manuscript. Please ensure that you refer to Table 1-7 in your text; if accepted, production will need this reference to link the reader to the Table.

Reviewers' comments:

Reviewer's Responses to Questions

**Comments to the Author**

1. Is the manuscript technically sound, and do the data support the conclusions?

Reviewer #1: Yes

Reviewer #2: Yes

2. Has the statistical analysis been performed appropriately and rigorously? 

Reviewer #1: Yes

Reviewer #2: Yes

3. Have the authors made all data underlying the findings in their manuscript fully available?

Reviewer #1: Yes

Reviewer #2: Yes

4. Is the manuscript presented in an intelligible fashion and written in standard English?

Reviewer #1: Yes

Reviewer #2: Yes

5. Review Comments to the Author

Reviewer #1: Authors suggested to address the following comments and suggestions when preparing the revised version:

= Abstract: section needs to be re-drafted to be self-contained means it has to clearly show the hypothesis, methodology, techniques and tools used, and the results obtained.

= Keywords: Authors suggested to update the keywords by selecting more relevant terms. Keywords play important role in the appearance of the manuscript in scholars search which will give it more hits and more citations.

= Introduction: authors advised to add one more paragraph at the end of the section to show the organization of the rest of the paper.

= Authors suggested to go through the following references and they MAY make use of them in updating the introduction and the related work sections:

- T. Chen, J. Blasco, J. Alzubi, and O. Alzubi “Intrusion Detection”. IET Publishing, Vol. 1, No. 1, pp. 1-9, 2014.

- Omar A. Alzubi, Jafar A. Alzubi, Ala’ M. Al-Zoubi, Mohammad A. Hassonah, Utku Kose, “An efficient malware detection approach with feature weighting based on Harris Hawks optimization” Cluster Computing Journal, 2021. https://doi.org/10.1007/s10586-021-03459-1

- Omar A. Alzubi, Jafar A. Alzubi, K. Shankar, Deepak Gupta, “Blockchain and artificial intelligence enabled privacy-preserving medical data transmission in Internet of Things” Transactions on Emerging Telecommunications Technologies, August 2020

- Jafar A. Alzubi, “Bipolar Fully Recurrent Deep Structured Neural Learning Based Attack Detection for Securing Industrial Sensor Networks” Transactions on Emerging Telecommunications Technologies, August 2020, https://doi.org/10.1002/ett.4069

= Conclusion: The conclusion should be abstracted so authors need to consider re-drafting it.

= Authors need to confirm that all acronyms are defined before being used for first time.

= Authors need to confirm that all mathematical notations are defined when being used for first time.

= Authors suggested to proofread the manuscript after addressing all comments to avoid any typo, grammatical, and lingual mistakes and errors.

Reviewer #2: The article is very well written with a robust methodology. I have a few points to add. First, in section 4, the libraries Pandas, Numpy, Matplotlib, and Sklearn must be cited correctly. For example, the pandas' publication can be found here: https://conference.scipy.org/proceedings/scipy2010/pdfs/mckinney.pdf. Second, the conclusion section needs to be expanded, pointing out what the other publications mentioned in section 3 could not achieve. Also, the authors could highlight their results and contribution to the literature

6. PLOS authors have the option to publish the peer review history of their article (what does this mean?). If published, this will include your full peer review and any attached files.

Reviewer #1: No

Reviewer #2: **Yes: **Valdecy Pereira

---

## [Author Response · Author response to Decision Letter 0]

11 Sep 2022

[Response to Reviewer 1’s Comments]

Comment 1: Abstract: section needs to be re-drafted to be self-contained means it has to clearly show the hypothesis, methodology, techniques and tools used, and the results obtained.

Response: Thank you for carefully reading the manuscript. The abstract has been edited to include all mentioned component in your comment.

Comment 2: Keywords: Authors suggested to update the keywords by selecting more relevant terms. Keywords play important role in the appearance of the manuscript in scholars search which will give it more hits and more citations.

Response: Thank you for carefully reading the manuscript. The word "adversary" has been added to the list of keywords. Additionally, we chose our research's keywords based on NIST standards, which are terms that are more relevant to our field.

Comment 3: Introduction: authors advised to add one more paragraph at the end of the section to show the organization of the rest of the paper.

Response: Thank you for your comment. We have updated the introduction and added the final paragraph to show the organization of the rest of the paper.

Comment 4: Authors suggested to go through the following references, and they MAY make use of them in updating the introduction and the related work sections:

Response: Thank you for your suggestion. The first study mentioned, which is more relevant to our study, is cited in the background section.

Comment 5: Conclusion: The conclusion should be abstracted so authors need to consider re-drafting it.

Response: Thank you for carefully reading the manuscript. Based on your comment, we have abstracted the conclusion to be more clearly .

Comment 6: Authors need to confirm that all acronyms are defined before being used for first time. Authors need to confirm that all mathematical notations are defined when being used for first time.

Response: Thank you for your comments. We have addressed the comments accordingly.

Comment 7: Authors suggested to proofread the manuscript after addressing all comments to avoid any typo, grammatical, and lingual mistakes and errors.

Response: We appreciate you taking the time to read the manuscript. The paper was meticulously checked and proofread to make sure the language is improved.

[Response to Reviewer 2’s Comments]

Comment 1: The article is very well written with a robust methodology. I have a few points to add. First, in section 4, the libraries Pandas, Numpy, Matplotlib, and Sklearn must be cited correctly.

Response: We appreciate you taking the time to read the manuscript and give your comments. We added the references for the following libraries: Pandas, Numpy, Matplotlib, and Sklearn as you mentioned.

Comment 2: The conclusion section needs to be expanded, pointing out what the other publications mentioned in section 3 could not achieve. Also, the authors could highlight their results and contribution to the literature.

Response: Thank you for your suggestions. We have updated the conclusion according to your comments. Moreover, we have highlighted the contributions at the end of literature.

Thank you very much for reviewing our article.

Thank you again for your precise time and valuable efforts spent in reviewing our manuscript as well as your helpful and positive comments.

Sincerely,

The Authors

---

## [Decision Letter · Decision Letter 1]

27 Sep 2022

Adversarial Attacks Against Supervised Machine Learning Based Network Intrusion Detection Systems

PONE-D-22-13930R1

Dear Dr. Alshahrani,

We’re pleased to inform you that your manuscript has been judged scientifically suitable for publication and will be formally accepted for publication once it meets all outstanding technical requirements.

Kind regards,

Omar A. Alzubi

Academic Editor

PLOS ONE

Additional Editor Comments (optional):

Reviewers' comments:

Reviewer's Responses to Questions

**Comments to the Author**

1. If the authors have adequately addressed your comments raised in a previous round of review and you feel that this manuscript is now acceptable for publication, you may indicate that here to bypass the “Comments to the Author” section, enter your conflict of interest statement in the “Confidential to Editor” section, and submit your "Accept" recommendation.

Reviewer #1: All comments have been addressed

Reviewer #2: All comments have been addressed

2. Is the manuscript technically sound, and do the data support the conclusions?

Reviewer #1: Yes

Reviewer #2: Yes

3. Has the statistical analysis been performed appropriately and rigorously? 

Reviewer #1: N/A

Reviewer #2: Yes

4. Have the authors made all data underlying the findings in their manuscript fully available?

Reviewer #1: Yes

Reviewer #2: Yes

5. Is the manuscript presented in an intelligible fashion and written in standard English?

Reviewer #1: Yes

Reviewer #2: Yes

6. Review Comments to the Author

Reviewer #1: Authors enhanced the manuscript quality by incorporating all reviewers comments and suggestions. I have no further comments to add.

Reviewer #2: (No Response)

7. PLOS authors have the option to publish the peer review history of their article (what does this mean?). If published, this will include your full peer review and any attached files.

Reviewer #1: No

Reviewer #2: **Yes: **Valdecy Pereira

---

## [Editor Report · Acceptance letter]

6 Oct 2022

PONE-D-22-13930R1 

Adversarial Attacks Against Supervised Machine Learning Based Network Intrusion Detection Systems 

Dear Dr. Alshahrani:

I'm pleased to inform you that your manuscript has been deemed suitable for publication in PLOS ONE. Congratulations! Your manuscript is now with our production department. 

Kind regards, 

on behalf of

Dr. Omar A. Alzubi 

Academic Editor

PLOS ONE